# Non-invasive suppression of essential tremor via phase-locked disruption of its temporal coherence

Sebastian R. Schreglmann [1,20], David Wang[2,3,20], Robert L. Peach [4,5,6,20], Junheng Li [5,6], Xu Zhang [7,8], Anna Latorre[1], Edward Rhodes [5,6], Emanuele Panella[9], Antonino M. Cassara [10], Edward S. Boyden[11,12,13,14,15,16,17], Mauricio Barahona [4], Sabato Santaniello [7,8], John Rothwell [1], Kailash P. Bhatia [1✉] & Nir Grossman[5,6,11,12,18,19✉]

Aberrant neural oscillations hallmark numerous brain disorders. Here, we first report a method to track the phase of neural oscillations in real-time via endpoint-corrected Hilbert transform (ecHT) that mitigates the characteristic Gibbs distortion. We then used ecHT to show that the aberrant neural oscillation that hallmarks essential tremor (ET) syndrome, the most common adult movement disorder, can be transiently suppressed via transcranial electrical stimulation of the cerebellum phase-locked to the tremor. The tremor suppression is sustained shortly after the end of the stimulation and can be phenomenologically predicted. Finally, we use feature-based statistical-learning and neurophysiological-modelling to show that the suppression of ET is mechanistically attributed to a disruption of the temporal coherence of the aberrant oscillations in the olivocerebellar loop, thus establishing its causal role. The suppression of aberrant neural oscillation via phase-locked driven disruption of temporal coherence may in the future represent a powerful neuromodulatory strategy to treat brain disorders.

[1] Institute of Neurology, Department of Clinical and Movement Neuroscience, Queen Square, University College London (UCL), London WC1N 3BG, UK. [2] Computer Science and Artificial Intelligence Laboratory, Massachussetts Institute of Technology (MIT), Cambridge, MA 02139, USA. [3] NuVu studio Inc, Cambridge, MA 02139, USA. [4] Department of Mathematics and EPSRC Centre for Mathematics of Precision Healthcare, Imperial College London, London SW7 2AZ, UK. [5] Department of Brain Sciences, Imperial College London, London W12 0HS, UK. [6] UK Dementia Research Institute (UK DRI) at Imperial College London, London W12 0NN, UK. [7] Biomedical Engineering Department, University of Connecticut, Storrs, CT 06269, USA. [8] CT Institute for the Brain and Cognitive Sciences, University of Connecticut, Storrs, CT 06269, USA. [9] Department of Physics, Imperial College London, London SW7 2AZ, UK. [10] IT'IS Foundation for Research on Information Technologies in Society, 8004 Zurich, Switzerland. [11] Department of Media Arts and Sciences, MIT, Cambridge, MA 02139, USA. [12] McGovern Institute for Brain Research, MIT, Cambridge, MA 02139, USA. [13] Howard Hughes Medical Institute, Cambridge, MA 02142, USA. [14] Department of Biological Engineering, MIT, Cambridge, MA 02139, USA. [15] Department of Brain and Cognitive Sciences, MIT, Cambridge, MA 02139, USA. [16] Centre for Neurobiological Engineering, MIT, Cambridge, MA 02139, USA. [17] Koch Institute for Integrative Cancer Research, MIT, Cambridge, MA 02139, USA. [18] Centre for Bio-Inspired Technology, Department of Electrical and Electronic Engineering, Imperial College London, London SW7 2AZ, UK. [19] Centre for Neurotechnology, Imperial College London, London SW7 2AZ, UK. [20] These authors contributed equally: Sebastian R. Schreglmann, David Wang, Robert L. Peach. ✉email: k.bhatia@ucl.ac.uk; nirg@ic.ac.uk

Synchronous oscillatory firing in large populations of neurons has diverse functional roles in the central nervous system (CNS), including regulation of global functional states, endowing connectivity during development, and providing spatiotemporal reference frames for processing of sensory input[1,2]. Aberrant synchronous oscillations have been associated with numerous brain disorders[3,4]. A palpable manifestation of such aberrant oscillation is pathological tremor in essential tremor (ET) syndrome, the most prevalent movement disorder affecting 0.4% of the general population[5]. While the biomolecular origin of ET remains elusive, rendering pharmacological interventions unspecific and often inefficient[6], its systems-level origin, i.e. oscillatory activity in the cortico-cerebello-thalamo-cortical (CCTC) network, is well established[7]. Invasive systems-level interventions such as lesioning and high-frequency deep brain stimulation (DBS) can successfully treat medication refractory ET[6,8], but their wide-scale application is limited due to the need for brain surgery. However, such aberrant oscillations fundamentally require a delicate cascade of coherent activities across the network components. We here explored whether such a cascade of coherent activities in the CCTC under ET can be disrupted non-invasively by perturbing the synchronous activity of the cerebellum via stimulation that is phase-locked to the tremor oscillation. To phase-lock the stimulation to the tremor oscillation, we first present a strategy to mitigate the Gibbs phenomenon distortion[9] from the Hilbert transformation[10] to compute the instantaneous phase of an oscillatory signal in real-time, a strategy that we called endpoint corrected Hilbert transform (ecHT). We then demonstrate that if transcranial alternating current stimulation (tACS) of the cerebellum is phase-locked to ET movement it can suppress its amplitude. Finally, we show that the suppression of ET amplitude is attributed to a disruption of the cascade of coherent activities in the olivocerebellar loop.

## Results

### Real-time computation of instantaneous phase via endpoint corrected Hilbert transform.

To enable phase-locking of stimulation to oscillatory activity, we first developed a strategy to compute in real-time the instantaneous phase of oscillatory signals. Traditionally, the instantaneous phase and envelope amplitude, of a band-limited, time-varying oscillatory signal are computed from a complexified version of the signal, known as the analytic signal, in which the real part is the unmodified signal and the imaginary part is the signal's Hilbert transform[10]. The discrete analytic signal is most accurately and efficiently computed in the frequency domain[11]. However, the Gibbs phenomenon[9] has made it impossible to accurately compute the instantaneous phase and amplitude at the ends of finite-length analytic signals[12]. We hypothesised that by applying a causal bandpass filter to the frequency domain representation of the analytic signal we would mitigate the Gibbs phenomenon by establishing a continuity between the two ends of the signal and remove the distortion selectively from the end part of the signal—aka ecHT. See Methods for a detailed description of the ecHT.

To assess whether the ecHT strategy could effectively mitigate the Gibbs phenomenon at the endpoint of the analytic signal, we computed the Hilbert transform of a test signal, i.e. a finite-length discrete cosine waveform, and quantified the error at the endpoint. Figure 1a, b show the Fourier spectra and the Hilbert transforms without the endpoint correction when the signal completed and did not complete full cycles within the sampled time interval, respectively. At the endpoint of the signal without ecHT, the maximal phase error was 179° (mean error 47 ± 50° standard deviation, st.d.), and the maximal amplitude error was 191% (76 ± 69%), Fig. 1c. Fig. 1d shows the same as Fig. 1b but

with the endpoint correction. At the endpoint, the ecHT strategy reduced the phase error by at least an order of magnitude (maximal error 12°; mean error 9 ± 2° st.d.) and the amplitude error by at least two orders of magnitude (8%; 4 ± 2%), Fig. 1e. The effects of the filter bandwidth and filter order are shown in Fig. 1f, g, respectively.

### Cerebellar stimulation phase-locked to essential tremor movement.

Next, we deployed the ecHT to test whether stimulation of the cerebellum phase-locked to the tremor movement can perturb ET in a cohort of 11 human participants with ET (see Supplementary Table 1 for demographic details). We measured the tremor movement of the hand, computed its instantaneous phase in real-time, generated eight different stimulating currents – sinusoidal at six different phase lags (0°, 60°, 120°, 180°, 240°, 300°), a control sinusoidal at the tremor frequency without phase-locking, and a sham, and applied them transcranially to the ipsilateral cerebellum via scalp electrodes (mean current amplitude 2.7 ± 1 st.d. mA). Fig. 2a shows a schematic of the phase-locked stimulation concept, Fig. 2b shows a schematic of the electrode configuration and the theoretical distribution of the electric fields in the brain, computed using finite element method (FEM) modelling. Supplementary Movie 1 shows a representative video. We applied each stimulation condition in a block of 60 s during which the participants maintained a tremor evoking posture. Each block consisted of a 30 s stimulation period (including 5 s of ramp-up and 5 s of ramp-down) and 15 s stimulation-free periods before and after. We repeated the stimulation conditions four times in a double-blinded random order with a 30 s rest interval between conditions and 5–10 min rest interval between sessions of eight stimulation conditions (see Fig. 2c for a schematic of the study design and Methods).

To assess whether the stimulating currents were delivered at accurate phase-lag, we computed, offline using Hilbert transform, the lag between the instantaneous phase of the stimulation waveforms and the instantaneous phase of the tremor movement waveforms. We found that during the phase-locked stimulation, the phase-lag distribution of each condition was narrow and different from the other conditions throughout the stimulation period (Fig. 2d(i)) and during the first and second halve periods (Fig. 2d(ii)), ($p < 10^{-8}$ for all periods; Fisher test; see Supplementary Table 2 for full statistics). The difference between the measured phase-lag and the set phase-lag was small, i.e. 3 ± 11° (mean ± st.d), across all the phase-locked conditions. The mean resultant vector length (quantifying the circular spread)[13], was close to one, i.e. 0.98 ± 0.01, across all the conditions, and did not differ between conditions throughout the stimulation period (Fig. 2e(i)), and during the first and second halve periods (Fig. 2e (ii); $p > 0.95$ for all periods; one-way ANOVA, see Supplementary Table 3 for full statistics). The mean resultant vector length was slightly larger at stimulation blocks with higher tremor amplitude (Fig. 2f(i)) and was slightly smaller at stimulation blocks with higher tremor amplitude st.d. (Fig. 2f(ii)) higher tremor frequency (Fig. 2g(i)) and higher tremor frequency st.d. (Fig. 2g (ii)). In contrast, during the sinusoidal stimulation without phase-locking, the phase-lag distribution was not different from a uniform distribution (Fig. 2d(i-ii); $p > 0.4$ for all periods; Omnibus test). The mean resultant vector length was small, i.e. 0.19 ± 0.071, and did not differ from sham stimulation ($p = 0.37$, paired Wilcoxon signed-rank test), indicating that the stimulation did not entrain the tremor phase (Fig. 2d, e and Supplementary Table 3). Across all stimulation conditions, the mean resultant vector length was not different in trials in which participants reported sensation underneath the electrodes and trials in which no sensation was reported ($p = 0.3$, Paired sign-rank test).

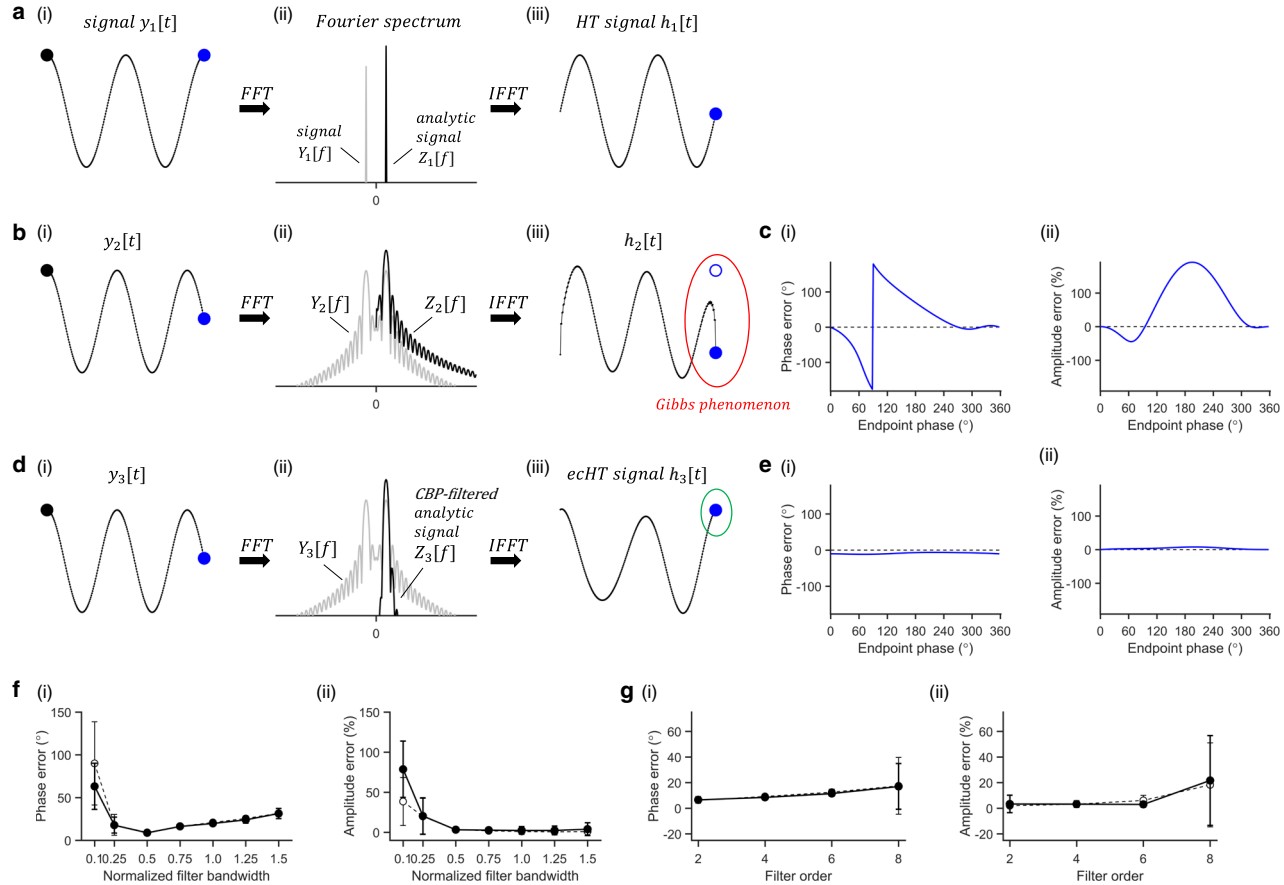

**Fig. 1 Concept and simulation of real-time computation of instantaneous phase and amplitude via ecHT. a** Hilbert transform (HT) of a finite, discrete, oscillatory signal completing full cycles. (i) Test signal $y_1$ in this example a cosine waveform with normalised amplitude, frequency $f_1 = 2Hz$, and phase delay $\emptyset_1 = 0$, sampled at 256 equidistant time-points over 1 s. First and last data points are marked with black and blue circles, respectively. (ii) Fourier spectrum (FS) $Y_1$, grey trace, of $y_1$, obtained via fast Fourier transform (FFT) of $y_1$ (in this example using 256 equidistant frequency-points), and FS $Z_1$, black trace, of the analytic signal, obtained from $Y_1$ by deleting the negative frequencies and doubling the amplitude of the positive frequencies; y-axis in log-scale. $Y_1$ trace at positive frequencies is overlaid by the $Z_1$ trace. (iii) HT $h_1$ obtained via inverse FFT of $Z_1$; filled blue circle, computed endpoint; non-filled blue circle, actual endpoint (in this case, overlaid by the filled circle). **b** HT of a finite, discrete, oscillatory signal not completing full cycles. Test signal $y_2$ similar to $y_1$ but with $f_2 = 2.25Hz$. Showing the same as in (**a**), but with FS sampled using 2048 points to illustrate the formation of the *sinc* waveform; red ellipse, outlines the Gibb phenomenon at the end of the signal. **c** Computation error of (i) phase and (ii) amplitude at the signal's endpoint for different end phases, simulated by varying $f_2$ between $2\,Hz$ and $3\,Hz$. **d** Endpoint corrected Hilbert transformation (ecHT) of the same signal in (**b**), i.e. $f_3=f_2$. Showing the same as in (**b**), but with the FS of the analytic signal multiplied by a response function of a causal bandpass (CBP) filter, in this example, 2nd order Butterworth bandpass filter with centre frequency $f_3$ and bandwidth $\frac{f_3}{2}$; green ellipse, outlines the mitigation of the Gibb phenomenon at the end of the signal. **e** Computation error of (i) phase and (ii) amplitude at the signal's endpoint obtained via ecHT. Showing the same as in (**c**). **f** Effect of filter's bandwidth on ecHT computation error of (i) phase and (ii) amplitude at the endpoint. Shown values are mean ± st.d.; $n = 180$ phase intervals between 0 and $2\pi$; filled black markers, error computed as in (**e**) for different filter bandwidths normalised to the filter centre frequency (in this example $f_3$); non-filled markers, error at the same data-point introduced by the filter, obtained by simulating a signal with a twice time interval to shift the Gibbs phenomenon from the original endpoint. **g** Effect of filter's order on ecHT computation error at the endpoint. Showing the same as in (**f**).

## Phase-dependent suppression of essential tremor amplitude.

After establishing that the stimulating currents were delivered at the desired phase lags, we assessed whether they affected the tremor amplitude. To quantify the stimulation effect relative to the baseline period and relative to the effect of sham stimulation, we computed, for each participant, the $z$-score of the tremor amplitude relative to the mean and the st.d. of the tremor amplitude during baseline in each stimulation condition, and then subtracted the median $z$-score of the tremor amplitude during sham stimulation (there was no significant difference in the tremor frequency and amplitude during baseline between conditions, see Supplementary Table 1 for full statistical details). To examine the temporal dynamics of the effect we quantified the $z$-score values during the first half and second half of the stimulation period, as well as during the post-stimulation period.

We found that the stimulation at the tremor frequency without phase-locking resulted in a tremor amplitude reduction, yet not statistically significant (Fig. 3a). A significant tremor amplitude change (reduction or increase) occurred in only a small number of participants (Fig. 3b and Supplementary Table 4). Across these subsets of participants, the change was statistically significant only in those showing a reduction and only during the first half of the stimulation (Fig. 3c, d). The corresponding percentage reduction during the first half period of the stimulation was $-10.8 \pm 3.0\%$ (mean ± st.d.) relative to baseline. In contrast, stimulation that was phase-locked to the tremor movement resulted in a significant reduction in the tremor amplitude, that increased throughout the stimulation period and sustained during the post-stimulation period (Fig. 3e; see Supplementary Fig. 1 for $z$-score values expressed relative to stimulation without phase-

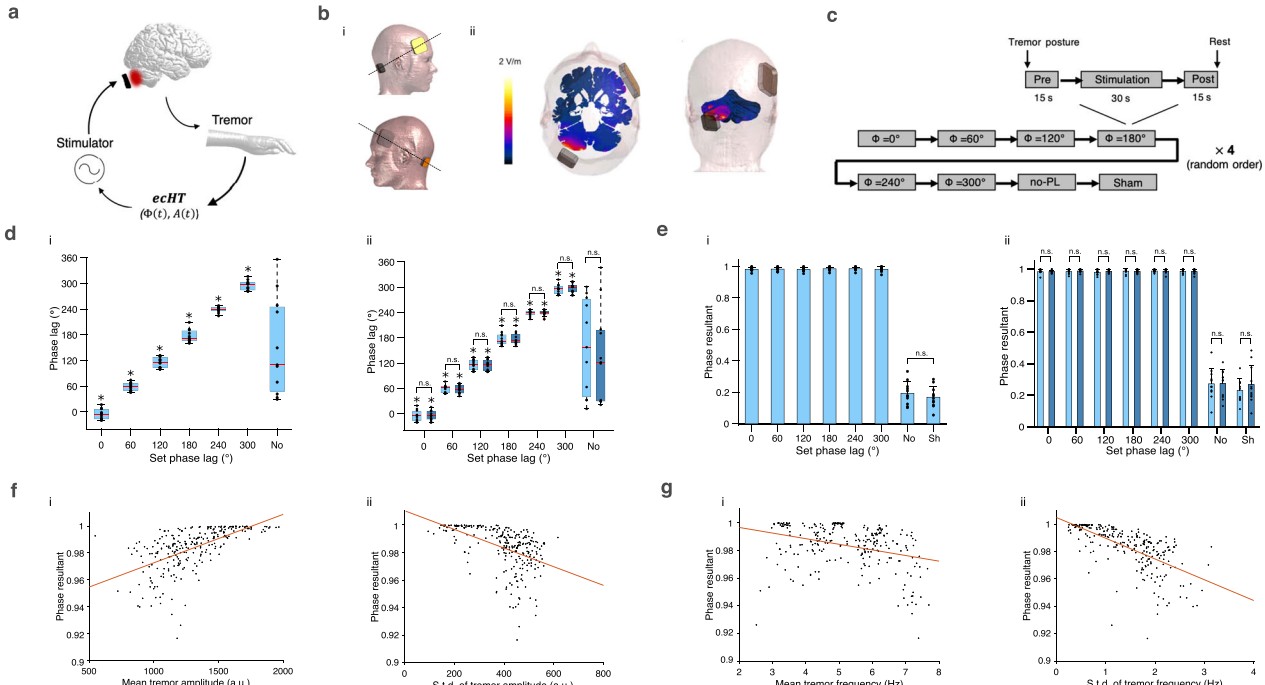

**Fig. 2 Stimulation of the cerebellum phase-locked to ET movement. a** Neuromodulation concept. ET is suppressed by perturbing its pathologic synchrony via cerebellar stimulation phase-locked to hand tremor oscillation. ET oscillation is measured via a motion sensor, instantaneous attributes of the oscillation (i.e. amplitude $A(t)$, phase $\Phi(t)$), are computed in real-time using ecHT, and electric currents are delivered, transcranially, to the cerebellum at a fixed phase lag. **b** Electrode configuration and cerebral electric fields distribution. (i) Stimulating currents were applied via a small skin electrode placed over the cerebellar hemisphere ipsilateral to measured hand tremor (10% axial nasion-inion distance lateral to inion) and a larger electrode placed over the contralateral frontal cortex (between F3 and F7 or F4 and F8 of the international 10–20 system). (ii) Finite element method (FEM) modelling of induced electric field for current amplitude of 2 mA. **c** Experimental design. **d** Phase-lag between stimulating currents and tremor movement vs. set phase lag during (i) whole stimulation period and (ii) 1st half (light blue) and 2nd half (dark blue) of the stimulation period. 'No', control sinusoidal current at the tremor frequency but without phase-locking; shown are box, 25 and 75% percentile values; horizontal red line, median value; horizontal black lines, data range; black markers, participants' values; *$p < 0.05$, two-sided Omnibus test; n.s. non-significant; $n = 11$ participants. See Supplementary Table 2 for between conditions statistics. **e** Mean phase resultant vector length vs. set phase lag during the same periods as in (**d**); shown are mean ± st.d.; markers show participants' values; 'No', stimulation with no phase locking; 'Sh', sham stimulation. two-sided ANOVA with post-hoc analysis using Wilcoxon signed-rank test; $n = 11$ participants; See Supplementary Table 3 for full statistics. **f** Mean phase resultant vector length vs. (i) tremor amplitude, (ii) st.d. tremor amplitude; shown black markers are trials' mean values. Red line, linear regression, (i) line slope $m = 0.59$, $p < 10^{-5}$, Pearson correlation test, (ii) $m = -0.49$, $p < 10^{-16}$. **g** Same as (**f**) but (i) tremor frequency, $m = -0.33$, $p < 10^{-7}$; (ii) st.d. tremor frequency, $m = -0.66$, $p < 10^{-32}$. Source data are provided as a Source Data file.

locking). The number of participants who showed a significant reduction in the tremor amplitude was significant during the second half of the stimulation and the post-stimulation period, while the number of participants who showed a significant increase in the tremor amplitude was not significant throughout (Fig. 3f and Supplementary Table 4; $p$ value threshold of amplitude change was Bonferroni corrected for six phase-locked conditions). Across these subsets of participants, the reduction/increase in the tremor amplitude was statistically significant throughout (Fig. 3g, h). The corresponding percentage reduction (and increase) in tremor amplitude during the first half period of the stimulation, second half period of the stimulation, and after the stimulation period, was $-18.1 \pm 2.5\%$ ($8.3 \pm 4.5\%$), $-15.2 \pm 2.2\%$ ($1.6 \pm 2.0\%$), and $-12.0 \pm 2.3\%$ ($6.5 \pm 3.3\%$), respectively, relative to baseline. The change in tremor amplitude was not different between sessions ($p = 0.64$, ANOVA; $p = 0.32$, linear mixed effect model with sessions as a predictor variable). Across all stimulation conditions, the z-score tremor amplitude was not different in trials in which participants reported sensation underneath the electrodes and trials in which no sensation was reported ($p = 0.54$, paired $t$-test).

Comparing the phase-locked conditions, we found that the reduction in tremor amplitude was close to significance (not

corrected) only at a phase-lag of 0° (Fig. 4a) but the number of participants who showed a significant reduction in tremor amplitude was not significant (Fig. 4b). However, if the phase lags of individual participants were expressed relative to the phase lag that resulted in the largest reduction of their tremor amplitude, the reduction in tremor amplitude and the number of participants who showed a significant reduction, were statistically significant, indicating a narrow range of efficacious phase that can vary between participants (Fig. 4c, d, see Supplementary Table 5 for complete statistical details). The corresponding percentage reduction during the second half period of the stimulation at 0° phase-lag was $-21.5 \pm 4.2\%$ relative to baseline.

To test whether the effect of the stimulation on the tremor amplitude is reproducible, we repeated the experiment in a subset of participants ($n = 6$, including participants 1,2,3,6, and 11 who showed a reduction in the tremor amplitude and participant 9 who did not; see Supplementary Table 1 for demographic and clinical details during the repeated experiment) and analysed the data in the same way as in the original experiment. We found that in the repetition experiment the stimulation currents were delivered at the same phase-lag accuracy as in the original experiment (Supplementary Table 6). As before, stimulation at

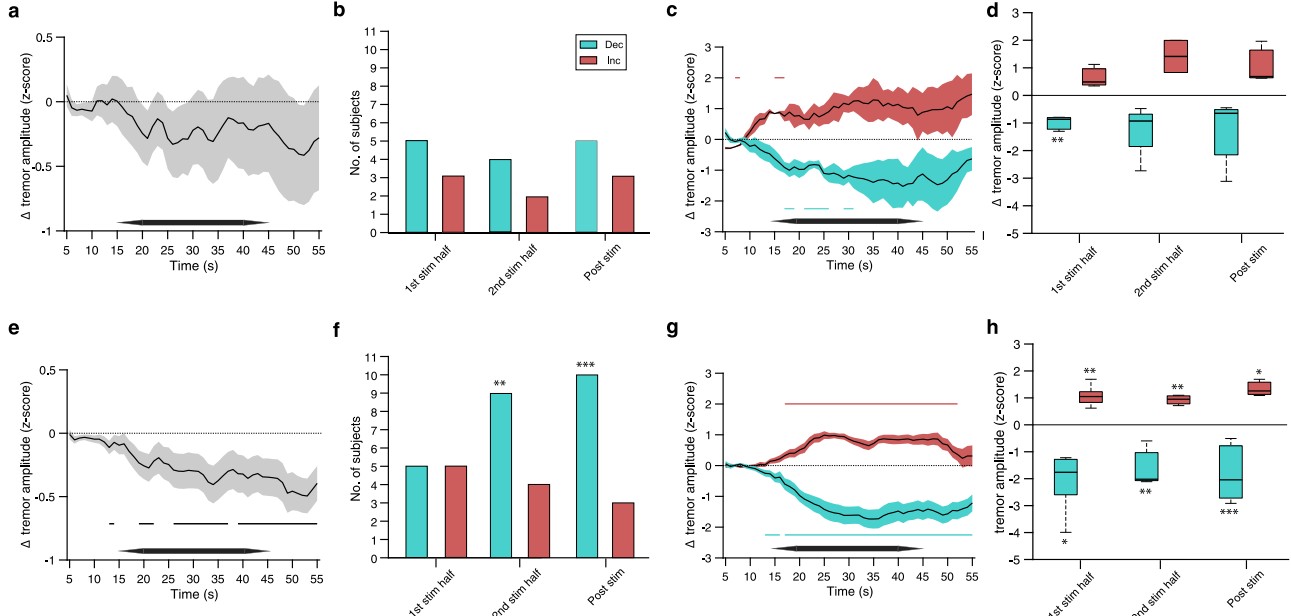

**Fig. 3 Characterisation of change in tremor amplitude induced by stimulation. a–d** Stimulating currents were applied at the tremor frequency but without phase-locking. **a** Change in tremor amplitude over time, shown are mean ± s.e.m. z-score computed using 10 s window every 1 s between 5 s and 55 s; horizontal black bar outlines stimulation period. **b** Number of participants with significant reduction (turquoise bars) and increase (red bars) in tremor amplitude during the first-half of stimulation period ('1st stim half'), second half of stimulation period ('2nd stim half'), and post-stimulation period ('post stim'); see Supplementary Table 4. **c** Change in tremor amplitude over time across participants with significant reduction (turquoise) and increase (red) in tremor amplitude during 2nd stim half in (**b**), shown are mean ± s.e.m. z-score; horizontal turquoise and red lines show corresponding epochs with significant z-score amplitude; horizontal black bar outlines stimulation period. **d** Change in tremor amplitude across the participants with significant reduction (turquoise) and increase (red) in tremor amplitude in (**b**), box plot shows 25 and 75% percentile values; horizontal red line, median value; horizontal black lines, data range, throughout the figure; from left-to-right $n = 5, 3, 4, 2, 5, 3$ participants. **e–l** Stimulating currents were phase-locked to the tremor movement. **e** Change in tremor amplitude over time, showing the same as in (**a**); horizontal black lines show epochs with significant z-score amplitude. **f** Number of participants with statistically significant reduction and increase in tremor amplitude in (**e**), showing the same as in (**b**); *, from left-to-right $p = 0.0019$, $p = 3.4 \cdot 10^{-5}$. **g** Change in tremor amplitude over time across participants with decreased and increased tremor amplitude during 2nd stim half in (**f**), showing the same as in (**c**). **h** Change in tremor amplitude in (**f**), showing the same as in (**d**); from left-to-right $n = 5, 5, 9, 4, 10, 3$ participants. Significance of z-score amplitude was analysed using unpaired two-sided t-test; Significance of number of participants was analysed using two-sided Fisher exact test against the number of participants who did not show a significant change; * indicates $p < 0.05$, **$p < 0.005$, ***$p < 0.0005$, n.s. non-significant throughout the figure. Source data are provided as a Source Data file.

the tremor frequency without phase-locking resulted in a tremor amplitude reduction, yet not statistically significant (Fig. 4e), however stimulation currents that were phase-locked to the tremor movement resulted in a significant reduction in the tremor amplitude that was sustained during the post-stimulation period (Fig. 4f). The participants who showed a significant reduction in the tremor amplitude during the stimulation period in the original experiment also showed a significant reduction in the tremor amplitude in the repetition experiment (see Supplementary Table 7 for full statistics). The z-score reduction in the tremor amplitude across those participants was not different from the original experiment (Fig. 4g). Comparing the phase-locked conditions, we found that across the cohort the reduction in the tremor amplitude was smaller at phase-lags of 0° and 300° (Fig. 4h, see also Supplementary Table 8 for full statistics). Within individual participants the phase-lag values that reduced the tremor amplitude were consistent in only 20% of the cases.

**Prediction of participants' response from distinct features of the tremor movement.** Next, we sought to explore whether the variability in the participants' response to the stimulation can be attributed to certain characteristics of their ET condition. We divided the participants into two groups, i.e. a 'responder' group ($n = 7$, including participants 1, 2, 3, 6, 8, 9, and 11) and a 'non-responder' group ($n = 4$, participants 4, 5, 7, and 10). A

participant was defined a 'responder' if his/her tremor amplitude decreased in at least one of the tested stimulation phases relative to sham and did not increase in any of the tested stimulation phases relative to sham, and a 'non-responders' if his/her tremor amplitude increased in at least one of the tested stimulation phases relative to sham or did not change in any of the tested stimulation phases relative to sham. We first assessed whether certain clinical or demographic characteristics can distinguish between responder and non-responder groups but found only non-significant trends of younger age ($p = 0.07$, Wilcoxon rank-sum test) and higher tremor frequency ($p = 0.08$) in responders (see Supplementary Table 1 for full statistical details). In addition, we did not find a difference between the groups in the amplitude of the applied currents ($p = 0.8$).

We then explored whether certain characteristics of the tremor movement can distinguish between the two groups. We deployed a feature-based statistical learning strategy[14] to extract 7873 different time-series features from a 10 s segment of the tremor movement before the onset of the stimulation in all the trials with phase-locked stimulation (301 trials in total, including 28 trials per participant except participant 3 in which only 21 trials were recorded); exemplary tremor traces are shown in Fig. 5a. We then used the features and a support vector machine (SVM) with a linear kernel to classify the tremor trials according to the subjects' responsiveness to a phase-locked stimulation. We found that

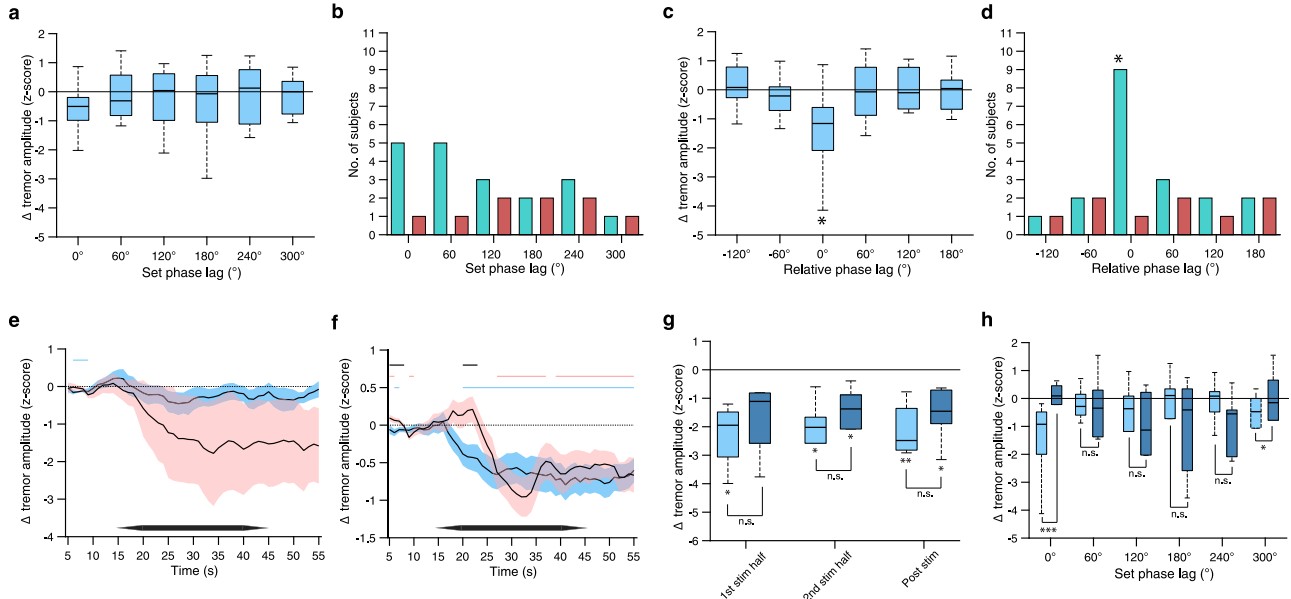

**Fig. 4 Characterisation of phasic dependency and reproducibility of induced change in tremor amplitude. a–d** Effect of the phase lag of stimulation. Shown values are for 2nd stim half. See Supplementary Table 5 for complete statistical data including 1st stim half and post-stimulation period. **a** Change in tremor amplitude vs. stimulation phase lag; $n = 11$ participants. **b** Number of participants with significant reduced (turquoise bars) and increased (red bars) tremor amplitude during 2nd stim half vs. stimulation phase lag. **c** Same as (**a**) but phase lags of each participant are expressed relative to the phase lag showing the largest reduction in tremor amplitude and wrap to ±180°. **d** Same as (**b**) but phase lags of each participant are expressed as in (**c**). **e–h** Characterisation of tremor amplitude during a repeated experiment in a subset of participants (participants 1,2,3,6, 9 and 11), see Supplementary Table 6 for statistics. **e** Change in tremor amplitude over time when stimulating currents were applied at the tremor frequency but without phase-locking, showing original experiment (blue) and repeated experiment (red); horizontal blue and red lines show epochs with significant z-score amplitude in original and repeated experiments, respectively; horizontal black lines show epochs with a significant difference in z-score amplitude between original and repeated experiments. **f** Same as (**e**) but stimulating currents were phase-locked to the tremor movement. **g** Change in tremor amplitude across the participants with significant in tremor amplitude in (**f**) in original experiment (light blue) and repeated experiment (dark blue); see Supplementary Table 7 for full statistics. **h** Change in tremor amplitude vs. stimulation phase lag, colour scheme as in (**g**); see Supplementary Table 8 for full statistics. Box plots throughout show 25 and 75% percentile values; horizontal red line, median value; horizontal black lines, data range. Significance of z-score amplitude and number of participants was analysed as in Fig. 3. Significance in (**c**) was also analysed using 2-sample Kolmogorov–Smirnov test. * indicates $p < 0.05$, **$p < 0.005$, ***$p < 0.0005$, n.s. non-significant throughout the figure. Source data are provided as a Source Data file.

using all the features, the tremor trials could be classified according to the participants' response with an accuracy of 97% (F-score of 96). However, even a small number of features was sufficient for high accuracy classification, using the top 1, 5, 10, and 40 features with highest single-feature classification accuracy, the tremor trials could be classified with an accuracy of 83%, 81%, 86%, and 92% (F-score of 82, 80, 85, and 91), respectively (Fig. 5b).

We then used a hierarchical cluster tree approach to search for the most informative features among the 40 features with the highest classification accuracy (Fig. 5c; feature values of individual participants did not differ between trials, $p > 0.5$; ANOVA). We identified 14 clusters of correlated features and extracted the corresponding features at the centre of those clusters—the list of the most informative features is given in Supplementary Table 9 and the magnitude probability density plots of exemplary features are shown in Fig. 5d (the classification accuracy plateaued at ~14 features, Fig. 5e). The extracted features revealed that the tremor movement in responders was smaller (Fig. 5dii), had a more sinusoidal like regularity (Fig. 5diii and Fig. 5div), and had a higher amplitude symmetry relative to zero (Fig. 5di). The Euclidean distance between feature centroids of the responders class and non-responders class was 0.55 (feature centroid of a class was computed by averaging the features across the corresponding samples). The feature centroids of individual participants who responded to the stimulation located at a distance <0.5 to the feature centroid of the responders class and

had a longer distance to the feature centroid of the non-responders class (exception was participant 8; Fig. 5f; distance of responders to responders' class, mean 0.35 ± 0.2 st.d.; responders to non-responders class, 0.6 ± 0.25; non-responders and responders class, 0.65 ± 0.15; non-responders and non-responders class, 0.35 ± 0.15).

To test whether these features of the tremor movement can potentially help to predict the response of participants to the stimulation, we repeated the experiment in a new cohort of seven human participants with ET. We analysed the data in the same way as in the original cohort and extracted the same 14 features from the 10 s tremor movement before the stimulation onset (see Supplementary Table 10 for demographic details, see Supplementary Table 11 for phase-locking and Supplementary Table 12 tremor amplitude statistics). We found that three participants (i.e. participants 2,3, and 7) responded to the stimulation based on the aforementioned responding criterion. The feature centroids of these participants, but not the rest of the cohort, were located at ≤0.5 distance to the feature centroid of the responders class from the original cohort and had a longer distance to the feature centroid of the non-responders class from that cohort (Fig. 5g) indicating a consistency in the relationship between the features of the tremor movement and the response to the stimulation.

**Suppression of essential tremor amplitude is underpinned by disruption of temporal coherence of movement.** After establishing that participants who responded to stimulation had

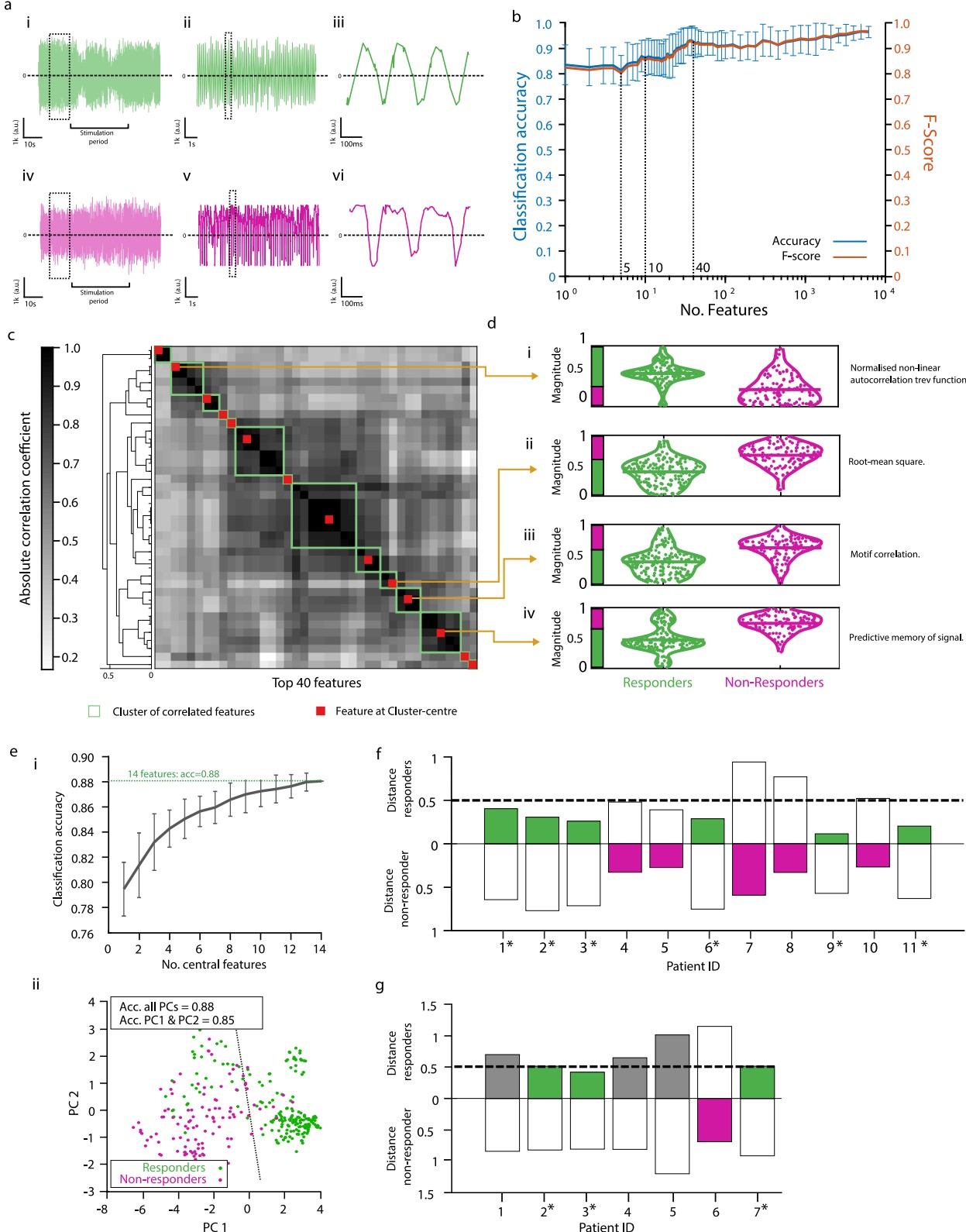

distinct characteristics of tremor movement during baseline, we next sought to explore whether the change in tremor amplitude during stimulation was associated with a change in other characteristics of tremor movement. We divided all the tremor trials with phase-locked stimulation (again 301 trials in total) into three datasets according to the change in tremor amplitude during stimulation relative to sham, i.e. trials with a decrease in tremor

amplitude ('decrease'; 58 trials from 11 subjects), trials with an increase in tremor amplitude ('increase'; 51 trials from 10 subjects; participant 6, did not show an increase in tremor amplitude in any phase-locked condition), and trials without a change in tremor amplitude ('no-change'; 192 trials from 11 subjects).

We then deployed the same feature-based statistical learning strategy[14] to test whether the characteristics of the tremor

**Fig. 5 Classification and prediction of participant's response via features extraction and statistical learning of the tremor movement. a** Exemplary recordings of tremor movement from a participant that showed a reduction in tremor amplitude during phase-locked stimulation relative to sham (i–iii) and one that did not (iv–vi). **b** Classification accuracy (blue) and *F*-score (orange) of participants' response as a function of the number of features. Shown are mean and st.d. values of the tenfold cross-validation. **c** Most informative features of the class structure. Shown are the 40 top predictive features in (**b**), clustered according to correlation coefficient and re-ordered according to the clustering; green box, outline of a feature cluster; red square, central feature of a cluster. See Supplementary Table 9 for a list of the features at the cluster's centre. **d** Normalised magnitude of exemplary features shown in (**c**) at the center of the clusters of correlated features. Green, 'responders' participants; magenta, 'non-responders' participants. See Supplementary Table 9 for description of the features. **e** Classification accuracy of participants' response using the 14 most informative features, i.e. the features shown in (**c**) at the centres of the clusters of correlated features, showing (i) mean classification accuracy ± st.d. vs. number of features, each repeated 100 times with a random selection of features out of the 14 most informative features, and (ii) 2D principal component analysis (PCA) plots of classification using all 14 features. Acc classification accuracy, PC principal component. **f** Euclidean distance between feature centroids of individual participants and the feature centroids of the responders' and non-responders' classes, using the 14 most informative features; *, indicates 'responders"; green bar, distance to responders class < 0.5 & distance to responders class < distance to non-responders class; magenta bar, distance to responders class > distance to non-responders class. **g** Same as (**f**) but for a new cohort of participants, showing distances to the same centroids of responders' and non-responders' classes in (**f**), i.e. of the original participants; grey bar, distance to responders class < 0.5 but distance to responders class > distance to non-responders class. Source data are provided as a Source Data file.

movement can distinguish between the stimulation and baseline periods in these three datasets. We extracted the same 7873 features as before from a 10 s segment of the tremor movement before the onset of the stimulation and from a corresponding 10 s segment during the middle of the stimulation; exemplary tremor traces with tremor amplitude 'decrease' and 'increase' are shown in Fig. 6a, b, respectively. We then used the features and the same SVM as before to classify the tremor trials according to the period class, i.e. 'baseline', or 'stimulation'. We found that the 'decrease' dataset had a higher probability of classification with high accuracy compared to the 'increase' and the 'no-change' datasets (Fig. 6c; 'decrease' vs. 'increase', $p = 0.01$; 'decrease' vs. 'no-change', $p = 0.008$; 'increase' vs. 'no-change', $p = 0.45$; and against a null distribution, generated by assigning random values to the feature), 'decrease', $p = 0.005$; 'increase', $p = 0.34$; 'no-change', $p = 0.58$; pairwise Kolmogorov–Smirnov test).

Focusing on the 'decrease' dataset, we found that using all the features, the tremor trials during stimulation and baseline could be classified with an accuracy of 79% (*F*-score of 79). However, the classification accuracy was dominated by only a few features, using the top 1, 5, 10, and 40 features with highest single-feature classification accuracy, the tremor trials could be classified with an accuracy of 78%, 79%, 79%, and 80% (*F*-score of 78, 81, 81, and 81, respectively; Fig. 6d). We then used, as before, the hierarchical cluster tree approach with a between feature correlation threshold of 0.2 to search for the most informative features among the 40 features with the highest classification accuracy (Fig. 6e). We identified nine clusters of correlated features and extracted the corresponding features at the centre of those clusters—the list of the most informative features is given in Supplementary Table 13 and the magnitude probability density plots of the central features with the highest probability are shown in Fig. 6f. We found that the classification was dominated by two time-series features, i.e. the 'information gain' feature, which estimates how easy it is to predict a data point in the time series from the preceding data points, and the 'quadratic fit of power spectrum cumulative sum' feature, which characterises the power spectrum of the time series. The increase in 'quadratic fit of power spectrum cumulative sum' during stimulation can be simply attributed to the drop in the spectral peak at the tremor's frequency. In contrast, the increase in 'information gain' during stimulation revealed a loss of linear dependency between consecutive data points of the tremor movement, i.e. a loss of temporal coherence.

To specifically test whether the change in the tremor amplitude was associated with a change in temporal coherence, we computed the change in the magnitude squared coherence during the stimulation period relative to the baseline period in

the 'decrease' and the 'increase' datasets as well as in a dataset consisting of all the trials with sham stimulation ('sham'). We found that the temporal coherence in the tremor frequency-band decreased in the 'decrease' dataset and increased in the 'increase' dataset during the stimulation, however, it did not change in the 'sham' dataset (Fig. 6g). The change in the tremor amplitude in the 'decrease' dataset, but not in the 'increase' dataset, was correlated with the change in the tremor temporal coherence. The change in the tremor amplitude in the 'sham' dataset was also positively correlated with the change in the tremor temporal coherence, however, with a smaller slope of the linear regression (Fig. 6h; combined dataset, line y-intercept $c = 0.2$, line slope $m = 1.2$, $R^2 = 0.32$; 'decrease' dataset, $c = -1.4$, $m = 1.35$, $R^2 = 0.49$; 'increase' dataset, $c = 0.94$, $m = 0.58$, $R^2 = 0.004$; 'sham' dataset, $c = -0.3$, $m = 0.78$, $R^2 = 0.32$; Pearson correlation; see Supplementary Fig. 2 for a correlation analysis of trials during stimulation without phase-locking). The change in temporal coherence in the 'decrease' dataset was correlated with the onset of the stimulation and was maintained during the duration of the stimulation (Fig. 6i).

To explore the possible mechanism by which the disruption of the temporal coherence could result in a suppression of the tremor amplitude, we simulated the CCTC network under ET condition[15] and phase-locked cerebellar stimulation. We found that the mechanism might be related to the suppression of the aberrant complex spikes in the Purkinje cells (PCs) of the cerebellum due to synchronisation of the hyperpolarizing phase of the stimulating with the onset of the complex spikes. See 'Neurophysiological model' in Supplementary Information.

## Discussion

In this paper we presented the ecHT strategy to compute the instantaneous phase of oscillatory signals in real-time and validated it using both simulation and measurements with pathologic oscillatory brain activity, i.e. ET. The ecHT strategy is based on the application of a causal bandpass filter to the DFT of the analytic signal to mitigate the distortion, known as the Gibbs phenomenon, from its end. Other frequency-domain and time-domain filters have been previously proposed to mitigate the Gibbs phenomenon from finite signals with a discontinuity[16] but these filters restore the DFT only away from the discontinuity itself[17]. There have also been reports of restoring the endpoint of the analytic signal using recursive models, such as autoregression[18] or polynomial fitting[19] to forward predict the physiological signal so that the last acquired data points are shifted from the window edge before the computation of the Hilbert transform. Recursive models have been recently tested for phase-locking brain stimulation[18,20,21], showing in some cases large st.d. (e.g.

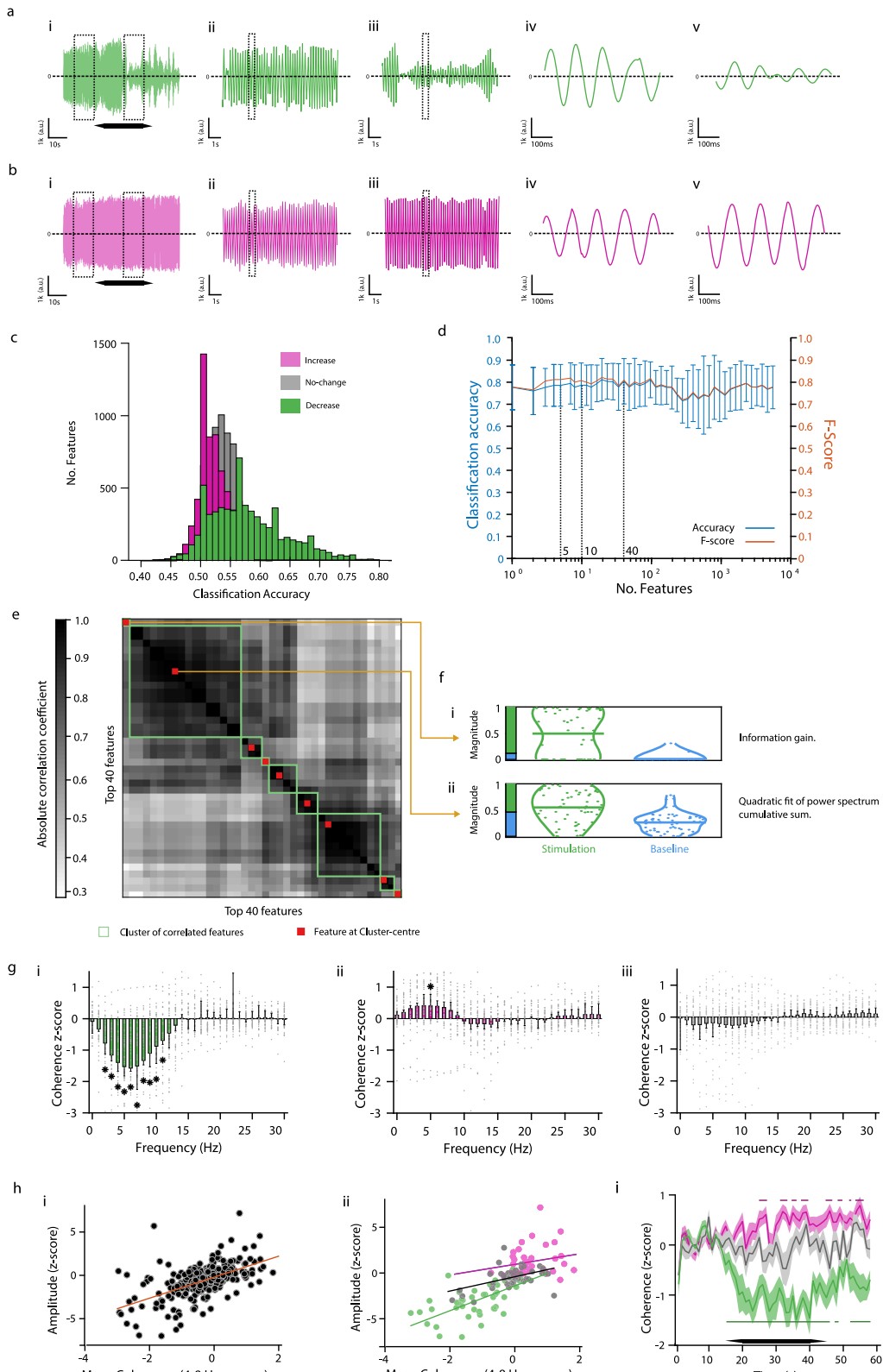

~54°)[21] and dependency on the coherence of the signal[18]. Ultimately, the high runtime complexity of recursive models (e.g. autoregression has a runtime complexity of $O(n^3)$ for $n$ samples, governed by the parameter estimation operation[22]) limit their use in applications that require real-time computation using conventional, and/or portable digital hardware.

In comparison, the ecHT is a simple, yet powerful method to accurately compute the Hilbert transform in real-time to track the instantaneous phase and envelope amplitude of an oscillatory signal. The ecHT maintains the same runtime complexity as the original Hilbert transform (i.e. $O(n\log(n))$ for $n$ samples), allowing implementation in simple and portable hardware. Future

**Fig. 6 Change in ET amplitude is linked to change in temporal coherence of the tremor movement. a** Exemplary recording of tremor movement during stimulation at a phase that resulted in a reduction of tremor amplitude relative to sham. (i) full 60 s recording; black hexagon, stimulation period. (ii) and (iii) magnified view of boxed region in (i); (iv) and (v) magnified view of boxed region in (ii) and (iii), respectively. **b** Exemplary recording of tremor movement from the same participant as in (**a**) but during stimulation at a phase that resulted in a small increase of tremor amplitude. (i–v) as in (**a**). **c** Probability distribution histogram of the feature-based classification accuracy according to the period class (i.e. 'baseline' and 'stimulation') of the 'decrease' (green), the 'increase' (magenta), and the 'no-change' (grey) datasets. two-sided pairwise Kolmogorov–Smirnov test. **d** Classification accuracy (blue) and $F$-score (orange) of the time-series traces in the 'decrease' dataset according to the period class (i.e. 'baseline' and 'stimulation') as a function of the number of features. Shown are mean and st.d. values of the tenfold cross-validation. **e** Most informative features for the class structure in the 'decrease' dataset. Shown are the 40 top predictive features in (**d**), clustered as in Fig. 5c. See Supplementary Table 13 for feature list. **f** Normalised magnitude of features shown in (**e**) at the centres of the clusters of correlated features. Green, 'stimulation' period; blue, 'baseline' period. See Supplementary Table 13 for feature description. **g** Change in tremor's temporal coherence. Shown values are mean ± st.d. $z$-score during stimulation relative to baseline period from (i) 'decrease' dataset (*, from left-to-right $p = 2.5 \cdot 10^{-6}$, $8.8 \cdot 10^{-8}$, $2.45 \cdot 10^{-8}$, $6.0 \cdot 10^{-7}$, $9.5 \cdot 10^{-6}$, $1.2 \cdot 10^{-5}$, $6.1 \cdot 10^{-6}$, $7.7 \cdot 10^{-6}$, $4.9 \cdot 10^{-5}$, $2.6 \cdot 10^{-4}$; $n = 49$ trials from 11 participants), (ii) 'increase' dataset (*$p = 0.0015$; $n = 41$ trials from 11 participants), and (iii) dataset of sham stimulation ('sham'; $n = 43$ trials from 11 participants); unpaired two-sided $t$-test with Bonferroni corrections for multiple comparisons of frequency-bins and datasets; grey markers, recording trails. **h** Correlation between change in tremor's amplitude and change in tremor's temporal coherence at the tremor frequency-band. (i) combined datasets and (ii) individual datasets with 'decrease', green; 'increase', magenta; 'sham', grey; each data point is a single trial **i** Change in tremor's temporal coherence at the tremor frequency-band over time. Shown values are mean ± st.d. with the same colour scheme as in (**hii**); horizontal lines show epochs with significant change; unpaired $t$-test with Bonferroni corrections for multiple comparisons of datasets; black hexagon, stimulation period. Source data are provided as a Source Data file.

studies may be able to improve the accuracy of the ecHT by adjusting, online, the central frequency of the bandpass filter to the instantaneous frequency of the signal, computed e.g. via a time derivative of the instantaneous phase. Given the widespread use of the Hilbert transform to compute the instantaneous attributes of oscillatory signals[10], the possibility for real-time computation using ecHT opens exciting opportunities in neuroscience and beyond (e.g. to monitor rotating engines and structural defects[23], speech analysis[24], and geophysics[25]).

We then used the ecHT to demonstrate the causal role of synchronous cerebellar activity in human participants with ET. By deploying for the first-time phase-locking stimulation to the cerebellum, we showed, in a double-blinded, sham and active controlled experiment, that ET amplitude can be efficiently supressed within a few seconds. The range of phases that were efficacious in suppressing the tremor in our stimulation was small but varied between participants and within participants between days of experiments perhaps due to differences in the electrode-skin capacitance. Future studies may be able to adjust the target stimulation phase online using similar closed-loop strategies currently deployed to adjust the target amplitude or frequency of DBS[26]. Our results exemplify the importance of accurate phase-locking to successfully induce a reduction in tremor amplitude. The fact that the tremor amplitude continued to drop during the stimulation period suggests that a longer stimulation period may yield an even larger suppression. The sustained drop in tremor amplitude after the end of the stimulation period may hold potential for a therapeutic effect via neural plasticity. To start testing the reproducibility of the stimulation effect, we validated the effect in a subset of participants a few years after the initial experiment and share the phase-locking methodology to allow other researchers to easily reproduce the experiment.

The rational of targeting the cerebellum in ET has been motivated by the recent discoveries of cerebellar abnormalities in ET patients and its strong connectivity to the basal ganglia (via the thalamic nuclei)[27]. Invasive phase-locked DBS of the thalamic Vim near the region receiving input from the cerebellum showed benefit in ET[28]. Nevertheless, numerous non-invasive cerebellar stimulation studies have failed to demonstrate a clear effect on ET severity even after multiple days of stimulation (see recent reviews[27,29,30]). For example, a prior study applying tACS to the cerebellum, but without phase-locking, found only a phase entrainment of the tremor with no effect on its amplitude[31]. There has been an original report that showed that non-invasive phase-locked stimulation of the motor cortex can ameliorate

tremor in Parkinson's disease (PD) patients[32]. Although both ET and PD are caused by aberrant oscillations in the motor system, their anatomical origins and degree of coupling between the central oscillators are very distinct[33]. Of course, the effect of stimulation on the activity of a brain circuit is complex, involving mixtures of local activation and inactivation pathways and interactions with downstream and upstream brain regions[34], and hence cannot be extrapolated across brain locations, brain states and diseases[27]. In fact, even a small change in stimulation parameters was shown to result in different and sometimes opposite effects[35,36] which may be particularly true in the case of the cerebellum given its both inhibitory and excitatory effects on the motor cortex[37,38]. There has also been a report that a periodic stimulation of the motor cortex at the tremor frequency without phase-locking, can entrain the phase of ET in patients undergoing DBS with an efficiency that was correlated to the somatosensory sensation underneath the electrodes[39]. In our study, the changes in the circular phase distribution and amplitude of the tremor were not dependent on the subjective sensation of the patients.

Finally, we showed, using data-driven statistical learning approach, that ET severity is linked to the temporal coherence of the movement, and that stimulation that disrupts the temporal coherence can reduce its severity. Hitherto investigations of the tremor coherence have focused on the correlation between two different tremor signals, such as the bilateral hand movement[40], intermuscular electromyography[41], and cortico-muscular[42]. These studies have elucidated important differences between diseases (e.g. ET vs. PD) however have not found a relationship to the severity of the tremor. The causal relationship between the amplitude of ET and its temporal coherence provides an important insight into the dynamics of the central oscillator underlying the disease. This is particularly interesting given the distinct relationship between the instantaneous frequency of ET and its fluctuation[43].

With almost a third of ET patients discontinuing medications due to insufficient benefit, medical contraindications, or the emergence of adverse effects[44], there is a pressing need for a novel treatment strategies for ET. Invasive DBS of the Vim nucleus is an alternative treatment for drug-refractory ET patients however, it is limited by the need for a brain surgery and the development of adverse side effects such as dysarthia and dysphagia[6,45,46]. Our results may provide the foundation for a new interventional strategy for ET. The mechanism of action of such an interventional strategy will be based on an active disruption of the cascade of coherent activities that generate the tremor oscillation in the

olivocerebellar loop. Our computational modelling suggests that it may be attributed to a timely perturbation of the generation of complex spikes in the PCs. Future computational studies may be able to explain the underlying mechanisms of those features predicting the stimulation outcome. Such a mechanism of action differs from the existing Vim DBS therapy for ET that masks the tremor oscillation in the thalamocortical (TC) loop but does not mitigate its generation in the olivocerebellar loop[15]. Future studies with larger patient cohorts and longer stimulation periods, are needed to better pinpoint the magnitude and duration of the tremor reduction and to assess the safety profile. In the future, neuromodulatory strategies that target the temporal coherence of the pathology may offer new opportunities to treat a wide range of brain disorders underpinned by aberrant synchronous oscillations.

## Methods

**Endpoint corrected Hilbert transform (ecHT).** A discrete analytic signal is most accurately and efficiently computed by deriving the discrete Fourier transform (DFT) of the signal, zeroing the Fourier components of the negative frequencies and doubling the ones of the positive frequencies, and constructing the analytic signal using the inverse discrete Fourier transform[11]. However, Gibbs phenomenon distortion[9] in the derivation of the analytic signal at the ends of finite-length signals has rendered an accurate computation of the instantaneous phase and envelope amplitude at the last data point impossible[12]. Since the Gibbs phenomenon stems from a nonuniform convergence of the DFT at a discontinuity between the beginning and the end of the analytic signal[47], we hypothesised that by applying a causal bandpass filter to the DFT of the analytic signal we would establish a continuity between the two ends of the signal and remove the distortion selectively from the end part of the signal. The bandpass feature of the filter reduces extraneous DFT coefficients, limiting the oscillatory properties to the target frequency-band, while balancing the phase-lag introduced by the low-pass component of the filter with the phase-lead introduced by the high-pass component of the filter. The causality feature of the filter restores the linear increment of the phase at the end of the analytic signal by projecting the oscillatory properties from the adjacent, non-distorted data points. Since the DFT treats finite sampled signals as if they were replicated periodically, the projection of the oscillatory properties would continue through the beginning of the signal, thus forcing a continued increment of the phase from the restored signal end to its beginning. The runtime complexity of the filtering is $O(n/2)$, where $n$ is the number of frequency points, is lower than $O(n \cdot log(n))$ of the fast Fourier transform (FFT) and inverse fast Fourier transform (IFFT) that dominates the computation of the analytical signal.

**Simulation of ecHT.** Simulation of ecHT was done in MATLAB (MathWorks Inc). A discrete oscillatory test signal

$$y_i[n] = A_i cos(2\pi f_i n - \emptyset_i) \quad (1)$$

was generated ($i$ being the signal number) over a finite time interval $T$, where $0 < n < N - 1$ was the time point number and $N$ was the total number of time samples, $A_i$ was the envelope amplitude of the signal, $f_i$ was the frequency of the signal, and $\emptyset_i$ was the phase delay of the signal. The analytic signal was computed by first computing the Fourier representation $Y_i[k]$ of the signal using MATLAB's fast FFT function ('fft'), where $0 < k < K - 1$ was the frequency bin number and $K$ was the total number of frequency samples. Then, generating the Fourier representation $Z_i[k]$ of the analytic signal by zeroing the Fourier components of the negative frequencies and doubling the Fourier components of the positive frequencies, i.e.

$$Z_i[k] = \begin{cases} Y_i & for\ k = 0, k = \frac{K}{2}, \\ 2Y_i[k] & for\ 1 \le k \le \frac{K}{2} - 1 \\ 0 & for\ \frac{K}{2} + 1 \le k \le K - 1 \end{cases} \quad (2)$$

If ecHT was applied, the Fourier representation of the analytic signal $Z_i[k]$ was multiplied with the response function $\sigma[k]$ of a Butterworth bandpass filter that was obtained using MATLAB's frequency response of digital filter function ('freqz') from the filter's impulse response coefficients generated using MATLAB's Butterworth filter design function ('butter'). Finally, the analytic signal $z_i[n]$ was computed from its Fourier representation $Z_i[k]$ using MATLAB's IFFT function ('ifft'). The phase of the signal at the last data point was computed via $atan\left(\frac{imag\{z_i[N]\}}{y_i[N]}\right)$, where $imag\{z_i[N]\}$ is the imaginary part of the analytic signal, i.e. the Hilbert transform of the original signal, and was compared to the actual phase of the signal at the last data point, i.e. $2\pi f_i N - \emptyset_i$. The amplitude of the signal at the last data point was computed via $\sqrt{imag\{z_i[N]\}^2 + y_i[N]^2}$ and was compared to the actual amplitude of the signal at the last data point, i.e. $A_i$.

**Feasibility study of cerebellar electrical stimulation phase-locked to ET**

*Ethics.* The study was approved by the local research ethics committee in accordance with the declaration of Helsinki. All participants provided written informed consent prior to study participation. Specifically, the study was approved by the Heath Research Authority (HRA; REC 03/N018, principal investigator John Rothwell, UCL). The approval included the assessment of governance and legal compliance, undertaken by HRA, with the independent Research Ethics Committee (REC) opinion provided by the National Hospital for Neurology and Neurosurgery and the UCL Institute of Neurology (ION) Joint REC. The overarching aim of the research project was to use transcranial brain stimulation paradigms to discover mechanisms of cortical excitability and their impact on motor behaviour. The research project was not classified as clinical trial or interventional trial by the HRA and hence did not required registration (which is mandatory for all clinical trials in the UK).

*Participants.* Eleven human participants with ET (three females) were recruited from the outpatient department of the UK National Hospital of Neurology and Neurosurgery, London. All participants fulfilled the diagnostic criteria for ET according to the Tremor Investigation Group and consensus statement of the Movement Disorder Society[48] and were on a stable treatment regime for their tremor for at least 30 days prior to the experiment. See Supplementary Table 1 for demographic and clinical information. Experiments were performed after overnight withdrawal of tremor medication during a single study visit in the dominant hand, or in case of slight asymmetry in the hand with the larger tremor amplitude. There were no drop-outs or adverse events noted.

*Participants (second cohort).* Seven human participants with ET (four females) were recruited as in the original to test whether their response can be predicted via the feature-based approach developed in the original study. See Supplementary Table 10 for demographic and clinical information. Experiments were performed as in the original cohort.

*Experiment design.* The experiment consisted of eight stimulation conditions, i.e. six sinusoidal stimulating currents that are phase-locked to the tremor movement at different phase lags (i.e. 0°, 60°, 120°, 180°, 240° and 300°), a control sinusoidal current at the tremor frequency but without phase-locking, and a sham stimulation condition. Each stimulation condition was applied in a block (i.e. trial) of 60 s during which the participants sat in an armchair and were instructed to maintain a tremor evoking posture, i.e. stretched, elevated arm with fingers parted, while their tremor movement was measured (see details below). The 60 s block included a 15 s of a baseline period, a 30 s of a stimulation period (including 5 s of ramp-up and 5 s of ramp-down at the beginning and end of the stimulation, respectively) and a 15 s of post-stimulation period. In sham stimulation blocks, the current was set to zero after the 5 s of ramp-up. Each 60 s block was preceded by a short (~4 s, 2048 data samples) calibration recording also in a tremor evoking posture to compute the tremor frequency and amplitude at the onset of the block (see details below). The eight stimulation conditions were applied consecutively with a 30 s rest interval between conditions. The sequence of eight stimulation conditions was repeated four times (apart from one participant in which they were applied three times due to fatigue) in a random order with 10 min rest period between sequences. The rest interval between conditions and the rest period between sequences were occasionally extended slightly if the participants requested.

*Measurement and real-time computation of instantaneous tremor phase via ecHT.* Tremor movements were measured using a 3-axis analog microelectromechanical system accelerometer (MMA7361, Freescale Semiconductor, Inc.; operated at a sensitivity range of ±1.5 G) that was attached to the proximal phalangeal segment of the middle finger using a custom-made adapter. The 3-axis acceleration measurements were sampled using three analog-to-digital converters of a microcontroller (Arduino Due with an Atmel AT91SAM3X8E processor and a single ARM Cortex M3 core; operated at a clock rate of 84 MHz) at a rate of ~500 Hz and an amplitude resolution of 12-bit, and the vector amplitude sum of the three axes was computed and stored in a running window of 128 samples. The instantaneous phase and amplitude of the tremor movement, i.e. at the last sample of the running window, were computed in real-time and at the same rate, using ecHT that was implemented on the microcontroller. The ecHT implementation had a 2nd order Butterworth bandpass filter (2nd order low pass, 2nd order high pass) with a bandwidth that was equal to half the frequency of the tremor and was centred at the frequency of the tremor. The frequency of the tremor was computed using FFT from a short calibration measurement of 2048 samples (i.e. frequency resolution of ~0.25 Hz) before each 60 s stimulation block. The sampled tremor movement measurement was logged to a laptop, together with the ecHT setting and the tremor frequency and amplitude computed during calibration, using a Processing script that was also used to interface with the microcontroller.

*Transcranial stimulation of ipsilateral cerebellum.* Sinusoidal stimulating currents were generated by first producing voltage waveforms, pseudo-differentially via two digital-to-analog converters of the microcontroller (with an amplitude range of ±1 V and an amplitude resolution of 12-bit) and then feeding them to an isolated biphasic current source (DS4, Digitimer Ltd; operated at an input range of ±1 V and

an output range of ±1 mA or ±10 mA). The frequency of each voltage waveform was equal to the frequency of the tremor computed before each 60 s stimulation block as mentioned above. To phase-lock a stimulating current to the ongoing tremor movement, the phase of the voltage waveform was adjusted, at the same rate of 500 Hz, to maintain a fixed phase lag to the computed phase of the last acceleration sample. The amplitude of the stimulating currents was 2.7 ± 1 mA (mean ± st.d.) across the participants, (the amplitude was individually adjusted for each participant below any discomfort level due to extraneous somatosensory stimulation underneath the electrodes). To reduce risk of extraneous high-frequency stimulation due to low signal-to-noise level, the amplitude of the voltage waveform was set to zero when the amplitude of the last acceleration sample was <1% of the amplitude during the short calibration measurement before each 60 s stimulation block. The generated stimulating voltage waveforms were logged to a laptop together with the tremor movement measurements using the same Processing script.

The stimulating currents were applied transcranially to the ipsilateral cerebellum via a $2 \times 2$ cm$^2$ skin electrode (Santamedical, $2'' \times 2''$ carbon electrode pad with Tyco gel that was cut to the specified dimensions) that was placed 10% nasion-inion distance lateral to inion (i.e. above the cerebellar lobule VIII) and was paired with a $5.08 \times 5.08$ cm$^2$ skin electrode (the same carbon electrode pad but was not cut) that was placed over the contralateral frontal cortex between F3–F7 or F4–F8 of the international 10–20 system. Before the placement of the electrodes, the scalp skin was prepared using 80% Isopropyl alcohol and an abrasive skin gel (NuPrep, Weaver and Company Inc), and a conductive paste (Ten20, Weaver and Company Inc) and/or a conductive gel (CG04 Saline base Signa gel, Parker Laboratories Inc) was deposited at the target locations. The resistance between the electrodes was maintained below 8 kOhm.

*Analysis of stimulation phase lag.* Analysis of the stimulation phase lag was done in MATLAB. The tremor movement trace of each 60 s block was filtered with the same filter settings that were used in the real-time computation, i.e. a 2nd order Butterworth bandpass filter with a bandwidth that was equal to half the frequency of the tremor and centered at the frequency of the tremor computed at the short calibration period preceding each block. The instantaneous phase of the stimulating waveform trace and the instantaneous phase of the filtered tremor movement trace were computed using MATLAB's 'hilbert' function, and the instantaneous phase lag between the two traces was calculated and then epoched in intervals of 1 s. The stimulating trace in the sham condition was a virtual sinusoidal waveform at the tremor frequency.

The statistics and statistical tests of the phase lag values were computed, using MATLAB's CircStat toolbox[13], in the following periods—the whole stimulation period (20 s since 5 s ramp-up time and the 5 s ramp-down time at the beginning and the end were excluded, respectively), the first half of the stimulation period (10 s since 5 s ramp-up time was excluded), the second half of the stimulation period (10 s since 5 s ramp-down time was excluded). First, the unimodality of the phase distribution of each stimulation condition was validated using Watson's test against a von Mises distribution (set phase 0°, $p < 10^{-5}$; 60°, $p < 10^{-5}$; 120°, $p < 10^{-5}$; 180°, $p < 10^{-5}$; 240°, $p < 10^{-5}$; 300°, $p < 10^{-5}$; no phase-lock, $p = 0.6$). The phase distribution during stimulation with phase-locking was not different from von Mises distribution but since the phase distribution during stimulation without phase-locking was different from von Mises distribution, we used non-parametric statistical tests. Next, the circular spread of the phase distribution of each stimulation condition was quantified by computing the length of the mean resultant vector $R$ and its uniformity was assessed using the Omnibus test. Then, the difference between the mean phase of the stimulation conditions was assessed using Fisher test and the difference between the mean resultant vector length $R$ of the stimulation conditions was assessed using ANOVA with post-hoc analysis using Wilcoxon signed-rank test. Finally, the effect of the tremor parameters, i.e. amplitude and frequency, on the length of the mean resultant vector $R$ was assessed via Pearson correlation.

*Analysis of change in tremor amplitude.* Analysis of the tremor amplitude was done in MATLAB. The tremor trace of each 60 s block was filtered as in the 'Analysis of stimulation phase lag'. The instantaneous amplitude was computed using MATLAB's 'hilbert' function and was epoched in intervals of 1 s. To express the tremor amplitude relative to the amplitude of the baseline period, the amplitude value of each epoch was z-scored by subtracting the mean value during the baseline period and then dividing by the st.d. of the value during the baseline period. The statistics and statistical tests of the tremor amplitude values were computed in the following periods—the baseline period (10 s between 3 s and 13 s from block onset), the whole stimulation period (as in 'Analysis of the stimulation phase lag'), the first half of the stimulation period (as in 'Analysis of the stimulation phase lag'), the second half of the stimulation period (as in 'Analysis of the stimulation phase lag'), and the post-stimulation period (10 s between 3 s and 13 s from stimulation offset). To assess the change in the tremor amplitude relative to the change in the tremor amplitude during the sham stimulation condition, the z-score amplitude values during stimulation and during post-stimulation periods of each stimulation condition were subtracted by the corresponding median z-score values of the sham stimulation condition.

To assess the effect of phase-locking the stimulation to the tremor movement, the change in the tremor amplitude due to stimulation with phase-locking and

without phase-locking was analysed. First, the change in the tremor amplitude due to each type of stimulation, i.e. without phase-locking and with phase-locking (data from all six phase-lags of stimulation was combined) was assessed across the participants in each epoch using unpaired *t*-test. Next, the change in tremor amplitude of individual participant due to each stimulation condition was assessed (i.e. data including four repetition trials from each phase-lag of stimulation was treated separately) during stimulation and post-stimulation periods using unpaired *t*-test as well as using surrogate distributions (i.e. 1000 z-scores values with the same st.d. but zero mean value), where the *p* value threshold of the stimulation conditions with phase-locking (but not without phase-locking) were Bonferroni corrected for the six phase lag conditions. Then, the number of participants that showed statistically significant increase/decrease of z-score amplitude was assessed using Fisher's exact test against the number of participants who did not show a change in the z-score tremor amplitude (participants could have a significant increase of z-score in one phase-lag and a significant decrease of z-score in another phase-lag). Finally, the z-score amplitude of the sub-group of subjects that showed a statistically significant increase/decrease of z-score amplitude was assessed using unpaired *t*-test.

To assess the effect of the phase lag value during stimulation, the change in the tremor amplitude due to stimulation with different phase lags was analysed. First, the change in the tremor amplitude due to each phase-lag of stimulation was assessed across the participants during the stimulation period using unpaired *t*-test. Next, the change in tremor amplitude of individual participant was assessed during stimulation again using unpaired *t*-test. Then, the number of participants that showed a statistically significant increase/decrease of z-score amplitude was assessed using Fisher's exact test. Finally, to account for differences in phase response across participants, the phase lags were expressed relative to the phase lag that resulted in the largest reduction in the tremor amplitude, and the change in tremor amplitude of individual participant and the number of participants with statistically significant change were reanalysed.

### Prediction of participants' response to stimulation from features of tremor movement

*Dataset.* Time-series of tremor movement during the baseline period, i.e. 10 s (5000 data points) from 5 s after the onset of tremor posture till 5 s before the onset of the phase-locked stimulation, were extracted from all the recorded trials with phase-locked stimulation, resulting in a dataset of 301 time-series trials (28 trials per participants except participant 3 in which only 21 time-series trials were recorded). The time-series were assigned a 'responder' or a 'non-responder' label if the participant responded or did not respond to the stimulation, respectively. A participant was conservatively labelled as a 'responder' if his/her tremor amplitude significantly decreased in at least one of the tested stimulation phases relative to sham and did not significantly increase in any of the tested stimulation phases relative to sham, and was labelled a 'non-responders' if his/her tremor amplitude significantly increased in at least one of the tested stimulation phases relative to sham or did not significantly change in any of the tested stimulation phases relative to sham.

*Extraction of time-series features.* For each time-series trace, 7873 features were computed using the highly comparative time-series analysis (*hctsa*)[14], resulting in a $301 \times 7873$ feature matrix. The computed features included autocorrelations, power spectra, wavelet decompositions, distributions, time-series models (e.g. Gaussian Processes, Hidden Markov model, autoregressive models), information-theoretic quantities (e.g. Sample Entropy, permutation entropy), non-linear measures (e.g. fractal scaling properties, nonlinear prediction errors) etc. All features with infinity or not a number (NaN) values and features with zero variance across the dataset were removed from the feature matrix, resulting in a reduced feature matrix of $301 \times 6196$. The value of each feature was individually normalised to the interval [0,1].

*Classification.* The feature space was partitioned, i.e. classified, using a linear SVM classifier, implemented with the *classify* function of MATLAB's *Statistics Toolbox*, which returned a threshold that optimally separated the two classes, i.e. 'responders' and 'non-responders' time-series. The accuracy of the classification was quantified by first computing the balanced classification accuracy $a = \frac{\text{precision} + \text{recall}}{2}$, and then computing the harmonic mean of precision and recall, i.e. $F_1$ score, $F_1 = \frac{2 \cdot \text{precision} \cdot \text{recall}}{\text{precision} + \text{recall}}$, where precision is the fraction of true positive classified samples over the total of positively classified samples and recall is the fraction of true positive classified samples over the total true positive and false negative classified samples. The classification was performed using a tenfold cross-validation to reduce bias and variance.

*Performance-based feature selection.* The univariate classification performance of each feature was evaluated against the class labels. A subset of 40 features with the highest single-feature classification accuracy was selected. To reduce the redundancy within the subset of features, the Pearson correlation distance, $d_{ij} = 1 - \rho_{ij}$ was computed for each pair of features, where $\rho_{ij}$ is the Pearson correlation coefficient between feature $i$ and feature $j$, and a hierarchical clustering was performed

using a complete linkage threshold of 0.2, resulting in clusters of features that were inter-correlated by $\rho_{ij} > 0.8$. The clusters of highly correlated features were then represented by the feature that was located most centrally within the cluster (i.e. at the cluster's centre).

*Feature-based prediction of participant response.* The centroid of individual participants in the feature space (including the extracted 14 most informative features) was computed by averaging the feature values across the corresponding trials. The centroid of the participant class (i.e. 'responders' or 'non-responders') in the same feature space was computed by averaging the features values across the corresponding trial dataset. The Euclidean distance between feature centroids was computed with *pdist* function of MATLAB.

*Visualisation using principal component analysis.* To facilitate visualisation of the feature space, principal component analysis (PCA) was performed. In this case, a covariance matrix was computed for the normalised set of features from which the eigenvectors and eigenvalues were extracted. Each principal component was constructed as a linear combination of the initial features. The first two principal components were then used to display 2D scatter plots of the features.

### Change in features of tremor movement due to stimulation

*Dataset.* Time-series of tremor movement during stimulation (10 s; 5000 data points; from 10 s after the onset of stimulation till 10 s before the offset of stimulation) and during baseline (10 s; 5000 data points; same as in 'Classification and prediction of participants' response to stimulation') from all trials with phase-locked stimulation (301 traces of stimulation and baseline each) were extracted and assigned a 'stimulation' class label or a 'baseline' class label, respectively. The 'stimulation' and 'baseline' time-series were then divided into three datasets according to the change in the tremor amplitude during stimulation, i.e. 'decrease', traces in which the tremor amplitude decreased during stimulation relative to sham (58 time-series of stimulation and baseline each, 11 subjects); 'increase', time-series in which the tremor amplitude increased during stimulation relative to sham (51 time-series of stimulation and baseline each, 10 subjects); 'no-change', traces in which the tremor amplitude did not change during stimulation relative to sham (192 time-series of stimulation and baseline each, 11 subjects). In addition, in a subset of the analysis, the same 'stimulation' and 'baseline' tremor traces were extracted from all the blocks with sham stimulation ('sham'; 43 time-series of stimulation and baseline each, 11 subjects).

*Extraction of time-series features, classification, and performance-based feature selection.* Same as in 'Classification and prediction of participants' response to stimulation'.

*Temporal coherence analysis.* The tremor temporal coherence vs. frequency of each tremor trace was quantified by computing the magnitude squared coherence across 1 s epochs during 'stimulation' period and 'baseline' period using MATLAB's *mscohere* function with a frequency range of 0–31 Hz and a 1 Hz frequency resolution. The computed values during 'stimulation' were then z-scored relative to the mean and st.d. of the values during 'baseline'. The tremor temporal coherence at the tremor frequency band was quantified by computing the mean z-score across the 4–8 Hz frequency bins. The tremor temporal coherence vs. time of each tremor trace was quantified by computing the magnitude squared coherence between 1 s epoch and its preceding one during 'stimulation' period and 'baseline' period using the same MATLAB's *mscohere* function, z-score the 'stimulation' values relative to 'baseline' in the same way, and then computing the mean z-score across the 4–8 Hz frequency bins. Statistical significance of magnitude squared coherence at a frequency bin was characterised for each dataset (i.e. decrease, 'increase', and 'sham') using unpaired *t*-test with Bonferroni corrections for multiple comparisons of frequency bins and datasets.

### Neurophysiological modelling

*Model description.* The CCTC network model under ET condition was simulated as in Zhang et al.[15]. The model is available on ModelDB (http://modeldb.yale.edu/266842). It consisted of 425 single-compartment, biophysics-based neurons from the olivocerebellar and TC loops, including 40 inferior olivary nucleus (ION) neurons in the brainstem, 200 PCs and 20 granular layer clusters (GrL; 3 distinct neurons per cluster, 60 neurons altogether) in the cerebellar cortex, 5 glutamatergic deep cerebellar projection neurons (DCNs) and 5 nucleoolivary (NO) neurons in the dentate nucleus, 5 ventral intermediate thalamus (Vim) TC neurons, 100 pyramidal neurons (PYN), and 10 fast-spiking interneurons (FSI). As in our previous study[15], the ET condition was simulated by reducing the conductivity and increasing the decay time of the PCs' GABAergic currents to the DCN, which mimics the loss of GABA$_A$ $\alpha_1$-receptor subunits and an up-regulation of $\alpha_2$/$\alpha_3$-receptor subunits in the cerebellum. Five instances of the model were considered and for each instance, simulations were repeated under normal condition, ET condition with no stimulation, and ET condition with stimulation of the cerebellum. Each simulation lasted 11,500 ms (integration step, 0.0125 ms). ET condition was initiated after 1000 ms and stimulation started after 1500 ms and lasted till the end of the simulation.

Hitherto computational studies of the effect of electrical stimulation on tremor activity have used a range of models ranging from a single cell with detailed biophysical and morphological representations[49] to thousands of cells in which their activity is represented by a simplified point-mass function[50], revealing complimentary insights. Neural network modelling has an inevitable trade-off between the scale and biological complexity of representation with both the size of the network and the biological complexity of individual cells affect the dynamics[51]. We chose to use a middle-ground approach with detailed biophysical representation but reduced morphological representation —an approach proven to be successful in the past by us[52] and others[53,54]. This approach may be particularly suited for ET since neural mass or mean-field models cannot represent the complex change in spiking pattern (rather than mean firing rate) observed in ET patients[55,56]. Furthermore, by maintaining a detailed biophysical representation of the cells, we could explore the effect of the stimulation on the interaction between the high-frequency simple spiking and low-frequency complex spiking of PCs that has been causally linked to ET[57].

To simulate the cerebellar stimulation, a current $I_{stim}$ was added to all the PCs in the model. $I_{stim}$ was sinusoidal with a frequency that is equal to the frequency of the ET and amplitudes between 1-5pA evoking small subthreshold depolarisations expected in our experiment. Specifically, $I_{stim}$ with an amplitude of 1pA induced a periodic depolarisation of ~0.5 mV amplitude in the single-compartment PC model which is similar to the depolarisation that was induced by an extracellular electric field with an amplitude of 2 V/m, predicted from our FEM modelling of the experiment (Fig. 2b), in the multi-compartment PC model (Supplementary Fig. 4a–b).

To validate that the direct response of the cerebellar cortex to the stimulating electric fields is dominated by the PCs, we simulated the response of the most abundant cell types in this region, i.e. PC and granule cell (GrC) to extracellular electric fields. To best capture the spatiotemporal dynamics, we used multi-compartmental models with detailed 3D geometrical reconstruction of the PC[58] and GrC[59]. We exposed the cells to homogenous extracellular electric fields that were aligned with the dendrite-somatic axes of the cells and induced the depolarisation. As in the original study with the PC model[58], we removed the sodium and calcium channels from the axonal initial segment of this cell to reduce its spontaneous pacemaker activity (see Supplementary Fig. 4a–b).

The amplitude of $I_{stim}$ was normalised to the average amplitude of the endogenous synaptic current to PCs, measured under ET state over 4000 ms (see also Perkel et al.)[60], with $I_{stim}$ of 1pA equals 4% of the average endogenous synaptic current to PCs. To phase lock the sinusoidal current to the ET oscillation, first the spike count trace of the TC neurons of the Vim was computed with a temporal resolution of 1 ms and then filtered using a 2nd order Butterworth bandpass filter with cut-off frequencies of 6 Hz and 10 Hz. Then, the instantaneous phase of the spike count trace was computed online every 10 ms using ecHT on a running window of 1000 ms, and the phase of the stimulating current was adjusted at those time-points to maintain the target phase lag.

*Computation of PCs phase-locking value.* The spike count trace of the PCs was computed with a temporal resolution of 1 ms (spikes were summed across PCs) and low pass filtered using a 2nd order Butterworth filter with a cut-off frequency of 30 Hz. Then, the instantaneous phases of the spike count trace and the stimulating current were computed offline using MATLAB's 'hilbert' function, and the instantaneous phase lag between the two was calculated every 1 ms. The phase-locking value of each PC was computed as in Lachaux et al.[61] and then averaged across the PCs.

*Computation of Vim power spectrum density.* First, the spike count trace of the TC neurons in the Vim was computed with a temporal resolution of 1 ms (spikes were summed across TC neurons). Then, the power spectral density (PSD) of the spike count trace was computed using Welch's method with 2000 ms Hanning window and 1000 ms overlap, and normalised to the total power between 0 Hz and 25 Hz. Tremor PSD was estimated as the peak PSD at the tremor frequency band, i.e. between 4 and 12 Hz.

*Computation of DCN and Vim temporal coherence.* The spike trains of the DCN and TC neurons of the Vim were low pass filtered using a 2nd order Butterworth filter with a cut-off frequency of 30 Hz, and the magnitudes squared coherence were computed using MATLAB's *mscohere* function with a frequency range of 0–30 Hz. Then the magnitude squared coherence in DCN and Vim during stimulation was expressed relative to baseline by subtracting the mean value during baseline and dividing by the st.d. value during baseline, i.e. z-score.

*Sensitivity analysis to the model size.* To explore the effect of the model size on the simulation outcome, we first repeated the simulation with a fivefold increase in the number of cells in the olivocerebellar circuit while keeping the other parts of the model unchanged, i.e. 'Model expansion 1'. Model expansion 1 consisted of 1425 cells, including 200 ION neurons, 1000 PCs and 20 GrL clusters (60 neurons altogether), 25 DCNs, 25 NO neurons, 5 Vim TC neurons, 100 PYN, and 10 FSI. We randomised the synaptic connections between the TC neurons and the DCNs with adjusted weights (20% of the original value) due to model expansion. Then, we repeated the simulation with a fivefold increase in the number of all cells in the

model i.e. 'Model expansion 2'. Model expansion 2 consisted of 2125 cells, including 200 ION neurons, 1000 PCs and 100 GrL clusters (300 neurons altogether), 25 DCNs, 25 NO neurons, 25 Vim TC neurons, 500 PYN, and 50 FSI. We randomised the synaptic connections between the different neuron types along the olivocerebellar circuit, and between TC neurons and DCNs, with adjusted weights (20% of the original value) due to model expansion.

**Transcranial electric field modelling.** Finite element method (FEM) electromagnetic simulations were performed in Sim4Life V.4 (ZMT ZurichMedTech AG, Zurich), using a quasi-static ohmic-current solver. Electrodes were created within the platform using Sim4Life's CAD functionalities and applied to the scalp of the MIDA anatomical head model[62]. Dirichlet (voltage) boundary conditions were assigned to the electrodes, and tissues electrical conductivities were assigned according to the IT'IS LF database[63]. A uniform rectilinear grid of 0.6 mm was used. The current between the electrodes was calculated integrating the current flux density on a closed surface surrounding one electrode and field magnitude were normalised to 2 mA input current.

**Reporting summary.** Further information on research design is available in the Nature Research Reporting Summary linked to this article.

## Data availability

Source data are provided with this paper. The tremor recording datasets used in this paper are available on the Harvard Dataverse repository https://doi.org/10.7910/DVN/Z6EN2I. Source data are provided with this paper.

## Code availability

The endpoint corrected Hilbert transform (ecHT) code implemented in Matlab is available as a supplementary file 'Supplementary_Code_1'. The highly comparative time-series analysis (hctsa) is available on GitHub https://github.com/benfulcher/hctsa. The Matlab code of the most informative features in Figs. 4 and 5 is also available as a supplementary file 'Supplementary_Code_2'. The NEURON model of CCTC network under ET condition and phase-locked electrical stimulation is available on the ModelDB repository http://modeldb.yale.edu/266842. The FEM model of the transcranial cerebellar electrical stimulation is available on the Harvard Dataverse repository https://doi.org/10.7910/DVN/H7RHQF.

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

## Acknowledgements

S.R.S. was supported by the Swiss National Science Foundation, Swiss Neurological Society and European Academy of Neurology and EMDO Foundation fellowship. X.Z. was supported in part by the CT Institute for the Brain and Cognitive Sciences IBRAiN Fellowship. S.S. was supported in part by the US NSF CAREER Award 1845348. K.P.B. was supported by Wellcome Trust MRC strategic neurodegenerative disease initiative award (WT089698), the Dystonia Coalition, Parkinson's UK (G-1009). M.B. was supported by EPSRC award EP/N014529/1 funding the EPSRC Centre for Mathematics of Precision Healthcare. E.S.B. acknowledges Lisa Yang, John Doerr, NIH R01MH117063, Edward and Kay Poitras, HHMI, and DARPA D20AC00004. N.G. was funded by the UK Dementia Research Institute (UK DRI)—an initiative funded by the Medical Research Council, Alzheimer's Society and Alzheimer's Research UK, Wellcome Trust fellowship (097443/Z/ 11/Z), Science & PINS Award for Neuromodulation, and NIHR IBRC Confident in Concept Award. The authors would like to thank Elisabeth Rounis and Tom Foltynie for helping with participant recruitment.

## Author contributions

S.R.S., designed and conducted clinical study, oversaw phase-locking, tremor amplitude and feature-based statistical learning analyses, wrote the paper. D.W., developed ecHT, design and implemented ecHT-based phase-locking brain stimulator. R.L.P., designed, developed and conducted feature-based statistical learning analysis. J.L., developed and conducted phase-locking and tremor amplitude analyses. E.R. and A.L., conducted the repeated clinical study. E.P., helped developing ecHT theory. A.M.C., conducted the FEM simulation. E.S.B., developed ecHT, oversaw experiments. M.B., designed and oversaw feature-based statistical learning analysis. X.Z. and S.S., designed, developed and conducted neurophysiological simulation of ET. K.P.B. and J.R., designed and oversaw clinical study. N.G., developed ecHT, designed and conducted clinical study, designed, developed and oversaw phase-locking and tremor amplitude analyses, oversaw feature-based statistical learning analysis and neurophysiological simulation of ET, and wrote the paper.

## Competing interests

N.G., D.W. and E.S.B. have applied for a patent on the ecHT technology, assigned to MIT, and founded a company that utilises it. K.P.B., received funding for travel from GlaxoSmithKline, Orion Corporation, Ipsen, and Merz Pharmaceuticals, LLC; serves on the editorial boards of Movement Disorders and Therapeutic Advances in Neurological Disorders; receives royalties from the publication of Oxford Specialist Handbook of Parkinson's Disease and Other Movement Disorders (Oxford University Press, 2008); received speaker honoraria from GlaxoSmithKline, Ipsen, Merz Pharmaceuticals, LLC, and Sun Pharmaceutical Industries Ltd.; personal compensation for scientific advisory board for GSK and Boehringer Ingelheim; received research support from Ipsen and from the Halley Stewart Trust through Dystonia Society UK. The rest of the authors declare no competing interests.
