## [Peer Review File · Nature Communications]

Editorial Note: This manuscript has been previously reviewed at another journal that is not operating a transparent peer review scheme. This document only contains reviewer comments and rebuttal letters for versions considered at Nature *Communications*.

REVIEWERS' COMMENTS

Reviewer #3 (Remarks to the Author):

As I stated in the my previous comments authors have addressed all my concerns related to the experiments and data analysis.

My only concerns left were about the computational model.

In the revised manuscript authors have addressed some of those concerns. But other remain. And this does not apply to just the arguments about the type of model authors have chosen. On that, I agree with the reviewers that while modelling we have to make some tradeoff.

The main issue is that whether the model is adding anything qualitatively to the manuscript either in terms of the insights or predictions. That is where I differ strongly with the authors. This conceptual different still remains. Indeed authors have added some predictions now. But I think the main contribution of the model would be to explain why the stimulation worked in only a subset of patients and other observations such as why it takes so many cycles to break the oscillations. These still remain unaddressed.

I do not want to delay the publication of the manuscript. Computational models is just an addendum which can be removed without any loss of importance of this work. But I am also fine it the model is pushed in the supp methods. My only objection would be to keep it in the main manuscript because (1) theoreticians will cringe about it and (2) for others there is a risk of creating misunderstanding.

Rebuttal Letter

The recent reviewers' comments to our manuscript submission are listed below in blue, point-by-point, together with our detailed response in black.

Reviewer #3 (Remarks to the Author):

As I stated in the my previous comments authors have addressed all my concerns related to the experiments and data analysis.

My only concerns left were about the computational model.

In the revised manuscript authors have addressed some of those concerns. But other remain. And this does not apply to just the arguments about the type of model authors have chosen. On that, I agree with the reviewers that while modelling we have to make some tradeoff.

The main issue is that whether the model is adding anything qualitatively to the manuscript either in terms of the insights or predictions. That is where I differ strongly with the authors. This conceptual different still remains. Indeed authors have added some predictions now. But I think the main contribution of the model would be to explain why the stimulation worked in only a subset of patients and other observations such as why it takes so many cycles to break the oscillations. These still remain unaddressed.

I do not want to delay the publication of the manuscript. Computational models is just an addendum which can be removed without any loss of importance of this work. But I am also fine if the model is pushed in the supp methods. My only objection would be to keep it in the main manuscript because (1) theoreticians will cringe about it and (2) for others there is a risk of creating misunderstanding.

We moved the model to the supplementary information as suggested by the reviewer. We refer to the supplementary model in the main text via the following sentences

“To explore the possible mechanism by which the disruption of the temporal coherence could result in a suppression of the tremor amplitude, we simulated the CCTC network under ET condition¹⁵ and phase-locked cerebellar stimulation. We found that the mechanism might be related to the suppression of the aberrant complex spikes in the Purkinje cells of the cerebellum due to synchronization of the hyperpolarizing phase of the stimulating with the onset of the complex spikes. See ‘Neurophysiological model’ in Supplementary Information.”